# Can co-created knowledge mobilisation interventions alter and enhance mindlines to improve childhood eczema care? A UK-based Social Impact Framework evaluation

Fiona Cowdell ,[1] Stephanie Lax,[2] Julie Van Onselen,[3] Rose Pendleton[4]

¹Faculty of Health, Education and Life Sciences, Birmingham City University, Birmingham, UK
²Centre of Evidence Based Dermatology, University of Nottingham, Nottingham, UK
³Independent Researcher, Oxford, UK
⁴Independent Researcher, Swadlincote, UK

**Correspondence to**
Professor Fiona Cowdell;
fiona.cowdell@bcu.ac.uk

## ABSTRACT

**Objective** To evaluate the impact of using knowledge mobilisation interventions to alter and enhance mindlines and improve childhood eczema care.

**Design** The eczema mindlines study involved three stages: (1) mapping and confirming eczema mindlines, (2) intervention development and delivery and (3) analysis of intervention impact. The focus of this paper is on stage 3. Data analysis was guided by the Social Impact Framework to address the questions: (1) what is the impact of this study on individuals and groups? (2) what changes in behaviour and practice have occurred due to their involvement? (3) what mechanisms have enabled these impacts or changes to occur? and (4) what are the recommendations and questions arising from this research?

**Settings** A deprived inner-city neighbourhood in central England and national/international settings.

**Participants** Patients, practitioners and wider community members exposed to the interventions locally, nationally and internationally.

**Results** Data revealed tangible multi-level, relational and intellectual impacts. Mechanisms supporting impact included: simplicity and consistency of messages adapted to audience, flexibility, opportunism and perseverance, personal interconnectivity and acknowledgement of emotion. Co-created knowledge mobilisation strategies to alter and enhance mindlines mediated through knowledge brokering were effective in producing tangible changes in eczema care practice and self-management and in 'mainstreaming' childhood eczema in positive way across communities. These changes cannot be directly attributed to the knowledge mobilisation interventions, however, the evidence points to the significant contribution made.

**Conclusion** Co-created knowledge mobilisation interventions offer a valuable method of altering and enhancing eczema mindlines across lay-practitioner-wider society boundaries. The Social Impact Framework provides comprehensive method of understanding and documenting the complex web of impact occurring as a result of knowledge mobilisation. This approach is transferable to managing other long-term conditions.

## STRENGTHS AND LIMITATIONS OF THIS STUDY

⇒ New, methodical application of the Social Impact Framework in the context of knowledge mobilisation.
⇒ Robust approach to mapping how knowledge mobilisation interventions have altered and enhanced eczema mindlines.
⇒ The Social Impact Framework offers a comprehensive approach to assessing contributions to changes in practice or behaviour, but definitive attribution claims cannot be made.

influence the stubborn evidence-practice gap in healthcare but measuring impact of these approaches is challenging. Childhood atopic eczema (AE) is a common and bothersome skin condition,[1] which requires regular and ongoing self-management.[2] AE is predominantly treated in primary care,[3] and a robust evidence base for treatment exists.[4 5] Effective self-management requires a level of shared knowledge, language and understanding between patient and practitioner.[6]

Co-methodologies in healthcare are widely considered to be a 'good thing' although the language of 'co' working is not fully defined and remains a fundamentally contested concept.[7] The terms co-design, co-production, co-creation, participatory research or participatory design are progressively used, sometimes interchangeably by researchers[8] and research funders.[9] Regardless of this, co-methodological working is gaining traction in healthcare,[10] and it is widely acknowledged that research engaging end-users is more likely to have an impact on practice.[11]

KMb interventions are increasingly used in healthcare to address multiple gaps between evidence, knowledge and action.[12] It requires purposeful efforts to create, disseminate and operationalise knowledge from multiple

## INTRODUCTION

Co-created knowledge mobilisation (KMb) interventions have the potential to

sources.[13] KMb is context specific,[14] relational[15] and socially constructed.[16] It is a rapidly evolving and wide-ranging field; currently, there are an excess of 47 models[17] and 71 published reviews[18] and a Google search yields 72 800 000 hits. Selecting approaches to KMb can be problematic, with some being highly theoretical and difficult to apply in practice. One pragmatic approach which is firmly embedded in day-to-day practice is alteration and enhancement of 'mindlines'. Mindlines are 'collectively reinforced, internalised tacit guidelines' which underpin clinical decision making,[19] particular emphasis is on contextual relevance and application of knowledge. Mindlines are developed from multiple knowledge sources such as communication with colleagues and opinion leaders and from personal tactic knowledge developed over time, knowledge is socially transmitted in the context of its use.[19] Mindlines build on the work of Nonaka and colleagues[20] who propose the Socialisation, Externalisation, Combination, Internalisation (SECI) spiral to guide implementation of new knowledge into practice. The SECI spiral comprises socialisation (surfacing tacit knowledge through shared experiences), externalisation (articulating tacit knowledge into explicit knowledge), combination (combining exposed explicit knowledge with more complex and systematic explicit knowledge, for example, clinical guidelines, to develop new knowledge) and internalisation (embodying this new knowledge as tacit knowledge for day-to-day use).

Impact of KMb is notoriously hard to measure. To date, the focus in healthcare has primarily been on moving new knowledge to clinicians and policy makers, with less attention paid to KMb across communities.[21 22] Effective evaluation of KMb activity is essential to better understand if and how stakeholders across communities use new knowledge and to refine strategies.[23] The Social Impact Framework (SIF), although primarily directed to evaluating co-production, offers a comprehensive and structured approach to understanding and documenting micro-meso-macro levels, processes, impacts and mechanisms of the KMb activity and to map the winding pathway of incremental and often subtle changes which are readily overlooked.[24] Beckett et al[12] provide a worked example of application of the SIF and their suggested questions are used here to guide analysis (box 1):

## METHODS

The eczema mindlines study involved three phases: stage 1: mapping and confirming eczema mindlines; stage 2: intervention development and delivery and stage 3: analysis of intervention impact (see figure 1). The focus of this paper is on phase 3, for context and clarity summaries of phases 1 and 2 are included.

### Phase 1: mapping and confirming eczema mindlines

Phase 1 comprised two elements. First, an ethnographic study to map lay and practitioner eczema mindlines in one deprived inner-city area in the UK.[25 26] Second, an

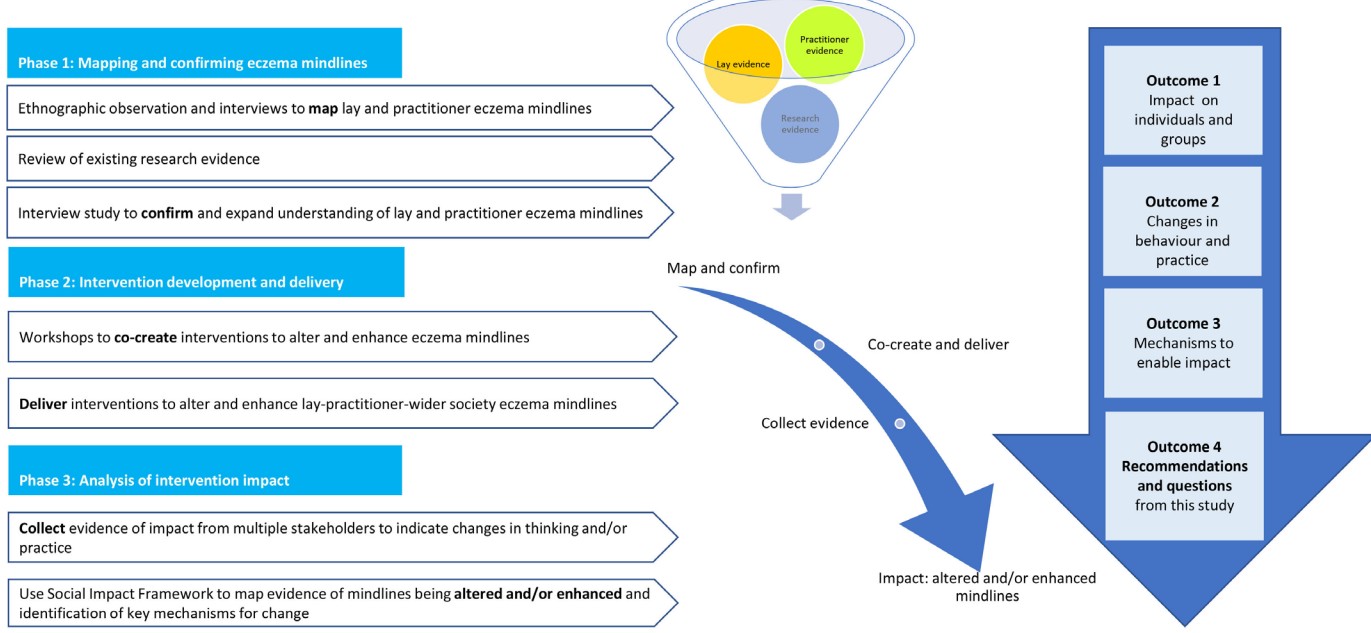

**Figure 1**   Evidence sources, stages and outcome measures (adapted from Beckett et al[12]).

---

**Box 2   Five co-created eczema messages**

1. Eczema is more than just dry skin.
2. Eczema does not just go away.
3. Moisturisers are for every day.
4. Steroid creams are okay when you need them.
5. You know your child's eczema best.

---

interview study with a wider population to confirm and expand understanding of lay and practitioner eczema mindlines.[27]

## Phase 2: intervention development and delivery

In a series of co-creation workshops involving people living with eczema, practitioners and researchers combined their tacit knowledge and data from phase 1 with existing research evidence. Co-creators concluded that to alter and enhance existing eczema mindlines five key, consistent, evidence-based messages needed to be shared (box 2).

Crucially, these messages needed to be transmitted across lay-practitioner-wider society boundaries using a range of techniques to enhance shared knowledge, understanding and language. For lay co-creators, 'trust' and 'realness' of messages was important and Health Care Practitioners (HCPs) (for example nurses, health visitors, community pharmacists and general practitioners (GPs)) wanted practical, locally relevant, hints and tips, tailored, 'no faff' approaches.[28] KMb interventions were developed in light of the characteristics of evidence and context in order to determine the best approaches.[29]

Intervention delivery was grounded in four schools of thought which cumulatively ensured knowledge was mobilised in the right format for the right audience and

---

**Box 3   Underpinning approaches to KMb**

Knowledge brokering
⇒ Knowledge brokers build networks and facilitate opportunities to share knowledge.[53] This was the core of all KMb activity.

'Ba'
⇒ 'Ba' is a shared space for knowledge generation and spreading[54–56] which aligns with the socialisation, externalisation, combination, internalisation spiral from which mindlines evolved. In this case, the Ba space was the locality of the original research, a deprived inner-city catchment. The intention was to achieve a density of KMb in a local area to support shared understandings and language across lay-practitioner-wider society boundaries.

Ripple effect model
⇒ Ripple effect was used to amplify the impact of each KMb action, so one event produces effects which spread and produce further effects.[57 58]

Social marketing
⇒ Social marketing goes beyond simply conveying knowledge widely and is intended to directly influence healthcare actions.[59] Here, emphasis was on community outreach.[60]

KMb, knowledge mobilisation.

---

effectively spread across boundaries as summarised in box 3.

Using the strategies outlined above, the five co-created messages were shared using multiple interventions as summarised in table 1 and figure 2. The role of the knowledge broker, initially FC, and later a wider group of people involved in the KMb interventions (eg, teachers) was pivotal. It involved working collaboratively with key stakeholders to enable transfer and exchange of knowledge across boundaries in different contexts[30] and using varied mechanisms (as discussed in Outcome 3). Messages were integrated into a children's book '*The Dragon in My Skin*' (hereafter Dragon) with associated animation, song and teacher resources. Dragon resources were endorsed by the National Eczema Society and the Royal College of Nursing to enhance confidence; the content was real and trustworthy as required by lay co-creators.

## Stage 3: analysis of intervention impact
### Aim
To systematically evaluate the impact of co-created KMb interventions in altering and enhancing mindlines and improving childhood eczema care.

### Design
SIF evaluation.

### Data collection
Data collection was multi-factorial, data sources and collection methods are summarised in table 2.

### Data analysis
Data analysis was guided by the SIF[12 24] to capture multi-level processes, impacts and key mechanisms of our KMb activities. We collated data from all sources including transcripts of audio-recorded interviews, feedback from online meetings and events, testimonials, email correspondence, researcher observation and conversations, online surveys and metrics. Many artefacts such as children's drawings were sent to us with a written description of the thought and emotion behind them from either the child or the teacher. Where no words were offered, we described pictures in words to try to capture their essence. The varied data could not and should not be separated from knowledge of the study design and the condition and creation of data.[31] The language and 'things' were inseparable,[32] and so were analysed together using the collated datasets. Two authors (FC and RP) iteratively read, thought and wrote about and discussed the data in its entirety. We then interrogated data to address the four social impact review questions (box 1).[33] Evaluation is reported according to the four outcomes around: impact on individuals and groups; changes in thinking, behaviour and practice; mechanisms enabling these impacts and finally recommendations and questions.

### Reflexivity
A reflexive stance was maintained for the duration of the study acknowledging both the complexities of

---

**Table 1** KMb interventions

| KMb materials | Recipients |
| --- | --- |
| Postcards and posters with key messages and supplementary information (figure 2) | HCPs in local area including GPs, GP trainees, practice nurses, health visitors, community pharmacists and pharmacy counter assistants<br>Displayed in local infant and primary schools, libraries, places of worship, GP practices, community pharmacies |
| Mindline informed educational sessions, led by DNS | Health visitors, community public health nurses (n=36), GPs (n=18), practice nurses (n=8) |
| Shopping centre pod for rapid consultations with two DNSs | Consultations with customers (n=94) |
| In person and online story reading and activity sessions in<br>▶ Nurseries and primary schools<br>▶ Places of worship | Children (n=86), teachers and teaching assistants (n=11)<br>Children and parents (n=~50) |
| Eczema mindlines website | Freely available online |
| *The Dragon in My Skin* book the-dragon-in-my-skin-132634726304040297.pdf (windows.net) (https://bcuassets.blob.core.windows.net/docs/the-dragon-in-my-skin-132634726304040297.pdf) | Freely available online<br>Hard copies distributed to primary schools (n=792) with links to all other Dragon resources |
| Dragon workshops | Children with eczema (n=10), their parents, an author and professional orchestra members |
| Dragon premiere | Children with eczema, their parents, professional orchestra members and invited guests with an interest in eczema including practitioners and members of eczema organisations (n=62) |
| Dragon book the-dragon-in-my-skin-132634726304040297.pdf (windows. net) (https://bcuassets.blob.core.windows.net/docs/the-dragon-in-my-skin-132634726304040297.pdf) | Freely available online<br>Hard copies distributed to primary schools (n=792) |
| Dragon Teacher resource pack tdims-workpack-v2-long-132693393982395364.pdf (windows.net) (https://bcuassets.blob.core.windows.net/docs/tdims-workpack-v2-long-132693393982395364.pdf) | Freely available online |
| Dragon Animation<br>The Dragon In My Skin - School of Health Sciences | Birmingham City University (bcu.ac.uk) (https://www.bcu.ac.uk/health-sciences/research/centre-for-social-care-health-and-related-research/research-projects/eczema-mindlines/the-dragon-in-my-skin) | Freely available online |
| Dragon translations | Will be freely available online |
| Eczema mindlines documentary https://youtu.be/C4d_yxvHVPk | Freely available online |
| DNS, dermatology nurse specialist; KMb, knowledge mobilisation. | |

the world and researcher entanglement with the fullness of the research process[34] and our preconceived understandings.

### Patient and public involvement

Lay people were involved in the development of the research question. They co-created the five key messages in a series of workshops, contributed to KMb planning and delivery and one representative is a co-author of this paper. All PPI activity was conducted in line with National Guidance.[35]

### RESULTS

Results are documented according to the four outcomes.

### Outcome 1: what is the impact of this study on individuals and groups?

Impact on individuals and groups was significant. Members of the co-creation group (n=22 lay people, practitioners and researcher) reported new understandings of eczema care from the 'other' perspective, 'conversations show how little lay people and HCPs understand each other's worlds and how interested they are in getting new insigh*ts*' (researcher). Participants demonstrated a new respect for the skills, knowledge and experiences of others and similarly gained a deeper understanding of the challenges and constraints of others. Co-creation enabled cross-fertilisation of ideas alongside a realisation of the power each person has to make a difference. Lay members found new ways of, and confidence in, communicating with practitioners and researchers.

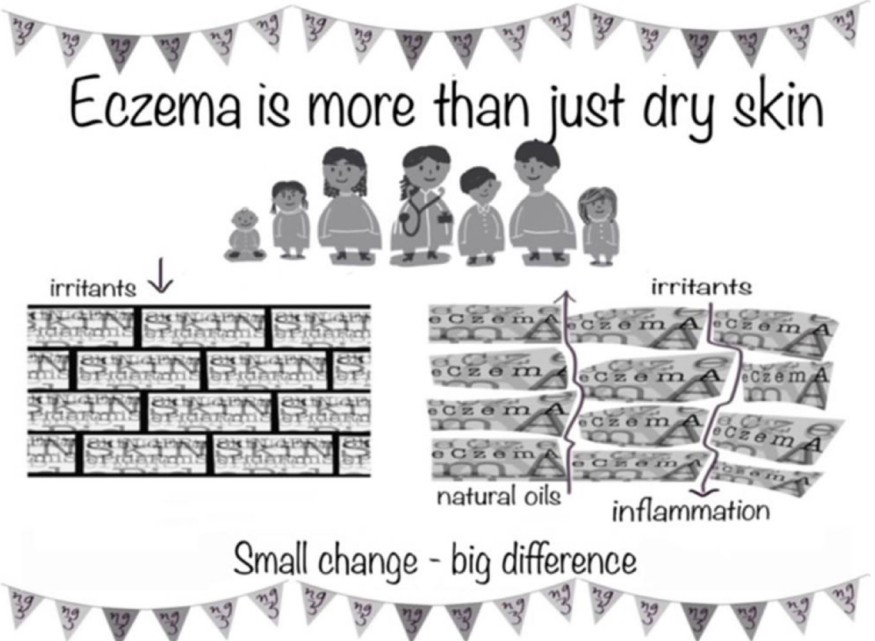

**Eczema is more than just dry skin**

irritants

natural oils

irritants

inflammation

Small change - big difference

BIRMINGHAM CITY University

**ECZEMA IS MORE THAN JUST DRY SKIN**

- 1 in 5 babies and children have eczema ('atopic eczema')
- Atopic eczema often runs in families
- There is no one cause

- A faulty skin barrier means the skin is dry, itchy and inflamed
- Eczema is made worse by 'triggers' which can be difficult to avoid

Information from Eczema Mindlines study, Birmingham City University

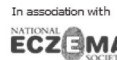
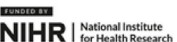

In association with NATIONAL ECZEMA SOCIETY

FUNDED BY NIHR | National Institute for Health Research

**Figure 2** Example of postcards with key messages and supplementary information.

At the time of writing engagement with the mindlines webpage, Facebook and Twitter are modest particularly given the vast potential audience numbers (box 4); this suggests that personal approaches may be more valuable.

Strategically placed posters and postcards impacted on the thinking of staff. Many either had or knew others who had eczema and thought the resources would be helpful. Pharmacy counter assistants supported adding a pack of postcards with all dispensed eczema topical treatments. A shopping centre pod set up for rapid consultations with two dermatology nurse specialists (DNSs) attracted 94 people in 1 day. Immediacy, advice from an expert and personalisation of the five key messages were highly valued in verbal feedback. It provided a basis for individuals to change their eczema self-management (although it is not possible to know whether this change was enacted). Children engaged enthusiastically with story reading and activity sessions, Teachers reported ongoing conversations about how it must feel to live with eczema and increased empathy both themselves and among children. One teacher reported "we had a lot of discussion around the topic of eczema and talked about feelings and how things such as eczema can affect our moods." Attendance at mindline informed eczema sessions for HCPs was higher than anticipated. Evaluation was overwhelmingly positive, particularly in terms of contextual relevance and applicability to own practice, for example "it stuck in my mind, direct relation to patients, the 'hook' to apply to own practice" (GP).

**Table 2** Summary of data sources and data collection methods

| Data source | Data collection method |
| --- | --- |
| HCPs including GPs, GP trainees, practice nurses, health visitors, community pharmacists and pharmacy counter assistants | Qualitative interviews, informal conversations, email feedback |
| Health visitor and community practitioners | Post session evaluation and subsequent email feedback |
| Children, parents and artists engaged in Dragon work | Observation, informal conversations, email feedback |
| Attendees at world premiere of Dragon event | Immediate comments, Zoom chat, follow-up emails |
| Teachers and student teachers | Online survey pre and post using Dragon resources in practice, informal conversations |
| Charitable organisations | Email, informal conversations |
| Professional organisations | Email feedback, testimonial |
| Social media | Metrics and testimonials |
| Researcher | Reflective diary of observation and experiences at all stages |

The Dragon book was described as "beautifully profound in its simplicity" (orchestra member) and "superb, I love the story and the pictures, what a lovely way for children with eczema to be able to see how they can tame their dragon and to have its impact validated in such a wonderful way!" (lay person). Through word-of-mouth connections around 200 further books were sent to other educators and HCPs. A dermatology specialist nurse shared the book with children attending her clinic and wrote "They love it, the children feel they have more control and it have made them feel special ………… this for me has been one of the best tools to use."

Children enjoyed the Dragon online co-creation sessions "[child's name] always looked forward to her sessions on zoom and you all made it so easy to engage and be confident. It's almost a shame the sessions are finished!". Parents commented "You guys do a great job at engaging the kids …… because you're after their input

---

**Box 4 Engagement with mindlines website**

⇒ All time views on the mindlines webpage to date 1146, split between documentary (n=386), animation (n=696) and knowledge nuggets (n=54).
⇒ Direct views to the video guides n=121. View quality was high, people spent on average of 5 min across all pages.
⇒ Uptake on Facebook and Twitter achieved 9 and 121 followers, respectively.
⇒ Animation views for the five messages were: (1) eczema does not just go away (n=270), (2) eczema is more than just dry skin (n=684), (3) moisturisers are for every day (n=58), (4) steroid creams are okay when you need them (n=223) and (5) you know your child's eczema best (n=23).

---

they're invested early on" and also valued validation of the realities of living with eczema. They were proud to be part of the online premiere. Feedback from this included

► "emotional…… this really helped me see what (my child) is feeling" (parent).
► One mother described her daughter's anger at having to manage the condition was struck "to hear that (anger) validated in a book for (my child) to understand."
► "It's beautiful. And as someone with a dragon since day one in life and struggling at the moment with it, I was especially moved by this." (teacher)
► "So often it's seen as 'just eczema'… it's nice to have something that shows how hard it is" (parent).
► "loved the way you haven't shied away from the difficult and painful experiences and feelings children have about eczema" (charitable organisation).
► "you've taken a debilitating but common and overlooked problem and made it come alive … I found it very moving" (HCP).

Parents noted the benefits of greater awareness of eczema among teachers and other children through widespread sharing of the Dragon resources "I am so glad that this will now be shared in schools to raise awareness amongst children of what some of their friends are going through" (parent), "Just awesome. That's so good. I'll send it to [child's name] teacher because there's a little girl in his class really suffering" (parent). The YouTube animation has been viewed 1378 times with 61.2% of views outside the UK, the average view duration was 3 min and 29 s and 22% of viewers watched all content relating to the five key messages (up to 9 min 45 s).

### Outcome 2: what changes in behaviour, practice and research have occurred due to their involvement?
#### Children and parents

Tangible changes in behaviour and practice were described. Some children and parents recounted more concordance with treatment, for example "[child's name] wanted me to tell you that she has put her spray on all over to look after her dragon" (parent). However, for most the more important change was the recognition that others gave to their child's eczema "so often it's seen as 'just eczema'… it's nice to have something that shows how hard it is" (parent).

#### Healthcare practitioners

Practitioners reported not necessarily learning anything new but rather 'fine-tuning' their mindlines and changing in thinking, for example, Dragon "validates experiences and feelings …… shows we understand" (HCP). Examples of simple but effective practice changes included:

► "the three main things that I took away from it were using one application of steroids is just as good as two. Go big early with the steroid and go greasier as well, really, rather than have a kind of hierarchy, going for a greasier emollient earlier rather than wait" (nurse) several months post-intervention the same nurse

reported "it's certainly made me more confident in prescribing, really. I don't think patients are coming back as much, I think actually going bigger earlier has a positive effect, really."

► "I think probably we've often a bit mean with it …. I've double checked that they've got enough of the emollients" (GP).

► "I readily use the information on my contacts at home visits and during clinic times, it is valuable to my practice and aids my prescribing for children with skin conditions, and when to refer" (HV).

► "I use (postcards) in practice and in teaching my students about care of the skin on a regular basis" (HV).

### Teachers

Use of resources led to more understanding from teachers and peers, "used it (book) with her individually to help the child manage her emotions and consider how she could manage her condition in school and that now the little girl picks the book up to read whenever she needs a 'comfort blanket' moment" (teacher). Teachers using Dragon resources conveyed value through words and images. "The children had some really mature discussion during this lesson and I have to say I was impressed, a couple of children with eczema were heavily involved in this and told other pupils some of their experiences (without being prompted or pressured to do so)" (teacher). "I have looked at this story with children in my class and they absolutely loved it. I created a hook where I had an animated dragon that came into our classroom and left some footprints and burnt paper and it got the children wondering why we had a dragon come in. The children then created their own story maps and understood the concept of eczema, as we have two girls who suffer from it. It was amazing", see figure 1. The implication from teacher feedback was that other children developed greater empathy for peers with eczema with the suggestion that this approach would reduce unkindness and bullying.

### Outcome 3: what mechanisms have enabled these impacts or changes to occur?

Multiple mechanisms enabled impact including: simplicity and consistency of messages adapted to audience, flexibility, opportunism and perseverance, personal interconnectivity and acknowledgement of emotion.

### Simplicity and consistency of messages adapted to audience

All KMb was underpinned by the five simple, key messages. Although not new these messages are at the heart of most eczema care with the mantra being 'get control-keep control' through use of topical corticosteroids when needed and regular and consistent application of emollients. Consistent, cross boundary messaging was intended to bring about shared language and understanding on which to base more equal eczema consultations. The role of the DNSs was pivotal. Mindline informed teaching engaged HCPs, using their own stories to 'hang things on' allowing immediate contextualisation and application of new knowledge. Equally having an expert with a wealth of current clinical and research knowledge and a repertoire of anecdotes made session rich, relevant and real. Lay people relished the opportunity to get on-the-spot expert, personal advice at the shopping centre. Immediacy was key to success. The five key messages provided a scaffold for each consultation, essentially each person received the same key information, but the DNSs, trusted sources of information, skilfully adapted and integrated messages to make them meaningful and useful to each individual.

The Dragon offered the five messages in child-friendly formats which addressed eczema care from a positive, proactive standpoint rather than the more usual problematisation of the condition. Dragon-related KMb activity has grown exponentially mainly by word of mouth supplemented by sharing in newsletters, magazines and websites. Numerous requests for resources have been received from HCPs and teachers. For example, one National organisation with a mission to transform localities with creativity and culture wrote the Dragon is 'Beautifully composed, created, animated and such a positive piece for children and young people to be involved in when eczema can be so hard' and went on to share the resources across wide-ranging networks. An attendee at the Dragon premiere descried the resources as 'incredible' and shared them with every primary school and primary care practice in one region. An education leader who heard about the Dragon through a personal contact wrote "I am delighted to be able to share these resources with our 87 mental health leads …. as I believe that this resource can support reducing the stigma linked to eczema, often born out of ignorance of the condition". Through HCP contacts Dragon resources are in the process of being translated and culturally adapted into French and Portuguese.

### Flexibility, opportunism and perseverance

Diligent, persistent, adaptable and proactive knowledge brokering was an essential element of enabling impact, as was perseverance in the face of practical and process constraints. Perseverance and patience were required in managing bureaucracy in setting up events and when events were cancelled at the last minute and needed to be rebooked. Some people rejected my offer outright including one children's play venue manager who would not support anything that suggested steroid creams were okay when you need them and a leisure centre manager who stated the messages were 'not suitable'. Effective knowledge brokering also relied on (1) building robust and enduring relationships with leading eczema charities and professional organisations and securing their endorsement, (2) engaging with influencers, authority figures and decision-makers and (3) openness to collaborative working across new networks. In the first instance, FC was the sole knowledge broker but over time others took on this role in different contexts (eg, teachers shared

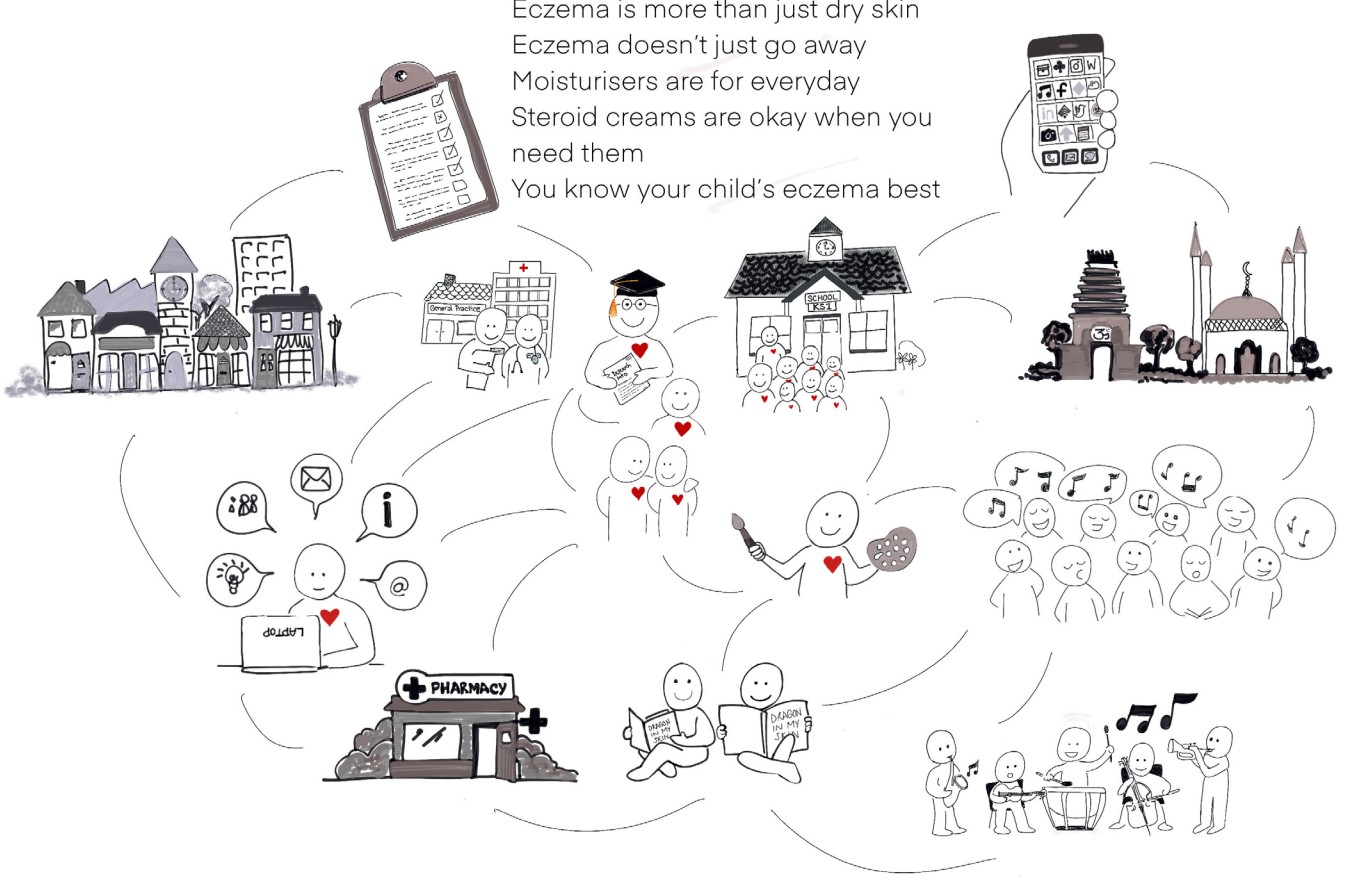

**Figure 3** Illustration of interconnectivity.

messages and resources far and wide), thus building up a knowledge sharing network.

## Personal interconnectivity

Personal interconnectivity was a key factor in sharing messages (figure 3). Through personal contacts, email and telephone calls, multiple individuals and organisations were contacted offering to share the key eczema messages using approaches tailored to each venue. Personal networks were effective door openers, for example, a practice manager introduced me to an Imam, who introduced me to a pharmacist and so the ripple went on, allowing me to access many groups I would not otherwise have reached. Equally, it was the starting point and central in developing, sharing and using Dragon resources. The idea was influenced by conversations with a patient group who highlighted the need to make teachers and children more aware of eczema and mainstream it rather than using existing problematising approaches such as having a special assembly on the condition. A chance conversation with a colleague led to development of the Dragon teacher resource pack. Consideration was given to the limitations of personal connections. We all inherently move in our own limited circles; however, we strived for inclusivity through situating our work 'out there' and using the ripple effect to meet new and unexpected allies.

## Acknowledgement of emotion

Tapping into emotions amplified the impact of KMb activity on altering and enhancing mindlines. For HCPs, relating knowledge to individual patients and their families was more powerful than generic teaching and sessions also gave space to express the frustrations of eczema care and collaboratively seek more positive approaches. For lay consultations, being 'listened to' as a whole person was key. Numerous Dragon comments focused on emotion as much as content, for example, a teacher wrote "It is such a wonderful concept that will make such a difference to children with and without eczema. I know me and my daughter would have felt much happier at school if we'd had something like this". An experienced HCP commented "You've taken a debilitating but common and overlooked problem and made it come alive! I loved it all and found it very moving" and many parents echo the sentiment of a charity leader "Loved the way you haven't shied away from the difficult and painful experiences and feelings children have about eczema".

## Outcome 4: what are the recommendations and questions arising from this research?

This research has important implications in terms of future KMb activity.

► First, altering and enhancing mindlines across patient-practitioner-wider society boundaries is possible and effective in changing behaviour/practice. Mindlines inherently made sense to all participants. Existing evidence was used to inform development of key, simple messages that were shared using creative and contextually adroit[19] formats that were relevant and applicable for end users.

► Second, knowledge brokering may start with one person but building up networks of knowledge brokers is essential. In this instance, the process was organic and was strengthened by openness to unexpected opportunities. In future, thought must be given to potential networks but equally researchers need to be open to and actively seeking new possibilities.

► Finally, the SIF offers a robust and iterative approach to planning, mapping and evidencing impact. 'Proving' the value of KMb is not and never will be straightforward. However, adoption of the SIF offers a step-change in demonstrating wide-ranging impact of KMb activity.

## DISCUSSION

The aim of this study was to evaluate the impact of using co-created KMb interventions to alter and enhance mindlines and improve childhood eczema care. It is one of the first to methodically evaluate the impact of using KMb interventions to alter and enhance mindlines across patient-practitioner-wider society boundaries. The evidence presented demonstrates the resonance that the work as a whole had with people living with eczema and those providing care. Recognition of the challenges and use of contextually relevant interventions for both appear to have increased receptivity and integration of new knowledge into everyday care. We are confident that eczema mindlines have been altered and enhanced. We have demonstrated that the SIF, which has a sound theoretical base, offers an effective and comprehensive approach to evaluating impact of KMb interventions. Use of the SIF has enabled reflection on the complex web of impact from a range of perspectives which may be overlooked if using more traditional measures. We are mindful that this work has limitations. We have made a contribution to changes in practice or behaviour but cannot definitively attribute this change to our interventions. However, the evidence presented suggests changes in people's thinking which is likely to influence their actions. Reporting is in accordance with the consolidated criteria for reporting qualitative research.[36]

Methodical assessment of the impact of KMb activity is scarce,[37 38] despite allied literature pointing to the need to build understanding[39] and competence[40] in this arena. Alternative approaches to evaluating KMb are available, for example, The Community Knowledge Mobilisation Framework[41], however, this is more limited than the SIF particularly in terms of considering breadth and mechanisms of change. Impact is a contested term, sometimes conceptualised as a linear process[23] in which impact is directly attributable to generation and dissemination of new knowledge.[42] In the present study, impact was viewed from the wide-ranging lens of the SIF. We are mindful that there are many other influences on eczema care and that this work offers a contribution to change.[43] Application of the SIF has allowed a nuanced understanding of the depth and breadth of impact of KMb activities and contributed to the much-needed development of KMb theory.[38] The SIF although primarily directed to evaluating co-production, offered a structured approach to reflect on micro-macro levels, processes, impacts and mechanisms of the KMb activity and map the winding pathway of incremental and often subtle changes which are readily overlooked.

The KMb interventions used to share simple consistent messages, co-created by end users are congruent with current thinking about challenges of KMb. Extant literature points to (1) information overload for HCPs[44] and lay people,[45] (2) inconsistent advice regarding eczema care,[2] (3) poor quality information and limited confidence in assessing veracity of available information for lay people,[46] (4) the need to consistently work with end users to increase uptake of knowledge[47] and (5) the value of promoting shared language and understandings and thus support shared decision making and self-management.[48] Gabbay and le May[19] identify the inter-relationship of patient-practitioner mindlines and hence the need to change mindlines in parallel. However, few studies have considered KMb across lay-practitioner-wider society boundaries.[21]

Knowledge brokers as intermediaries between researchers and practitioners are well established in healthcare as evidenced in recent reviews.[49 50] Nevertheless, the role can be problematic with some brokers challenged by role ambiguity and the need for a multi-dimensional skill set.[51] In the present study, the broker being a researcher and nurse and having lived experience of eczema minimised these tensions and were of distinct benefit in the relationship brokering component of the role.[52] Over time others took up brokering activity, which enhanced capacity to move evidence to practice.[51]

Systematic analysis of KMb activity has highlighted multiple mechanisms influencing impact which may be applied in future KMb work. In the present study, key processes included: (1 engagement of key stakeholders and end users; (2) appreciative engagement, creating opportunities for engagement, valuing unique individual contributions and respectful working; (3) diligent, persistent and proactive knowledge brokering; (4) sustained supportive relationships; (5) use of iterative flexible processes, adjustment to contextual challenges and changing circumstances and (6) creativity and use of diverse media. The KMb materials provide lay people, HCPs and teachers with evidence-based resources to use and share with others. We also offer

a novel approach to systematically evaluating KMb activity which builds much needed theory alongside practical application. There is still much work to be done to better understand the impact of knowledge mobilisation strategies specifically those striving to bridge lay-practitioner-wider society boundaries to improve care.

## Conclusion

This study is one of the first to systematically assess the impact of KMb interventions designed to alter and enhance mindlines across lay-practitioner-wider society boundaries. The SIF has been used to transparently map the complex web of impact from a range of perspectives which may be overlooked if using more traditional measures. Crucially, impact has included tangible changes in childhood eczema care practice and self-management and 'mainstreamed' the condition to enhance understanding of children and teachers. It brings to the fore new understandings of key mechanisms underpinning effective KMb practice. The challenge now is to test this approach to assess the impact of other types of KMb interventions.

**Acknowledgements**  We extend our thanks to all who have taken part in this study and to Kate Beckett who has kindly shared her knowledge and expertise when commenting on drafts of this manuscript. Thanks also to Jay Nolan Latchford and Michaela Tait for creating images.

**Contributors**  FC led the research, including study design, data acquisition and interpretation and writing this article. She is accountable for all aspects of the work in ensuring that questions related to the accuracy or integrity of any part of the work are appropriately investigated and resolved. SL contributed to study design, critical revision and gave approval for submission. RP contributed to study design, data acquisition and interpretation, critical revision and gave approval for submission. JVO contributed to data acquisition, critical revision and gave approval for submission. FC is the guarantor.

**Funding**  This report is independent research arising from a Knowledge Mobilisation Research Fellowship, Professor Fiona Cowdell, KMRF-2015-04-004 supported by the National Institute for Health Research.

**Disclaimer**  The views expressed in this publication are those of the authors and not necessarily those of the NHS, the National Institute for Health Research, Health Education England or the Department of Health.

**Competing interests**  None declared.

**Patient and public involvement**  Patients and/or the public were involved in the design, or conduct, or reporting, or dissemination plans of this research. Refer to the Methods section for further details.

**Patient consent for publication**  Not applicable.

**Ethics approval**  Not applicable.

**Provenance and peer review**  Not commissioned; externally peer reviewed.

**Data availability statement**  Data are available upon reasonable request. Data are available upon reasonable request to the corresponding author.

**ORCID iD**
Fiona Cowdell http://orcid.org/0000-0002-9355-8059

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
