## [Reviewer comments · BMJ Open]

ARTICLE DETAILS

TITLE (PROVISIONAL)	Can co-created knowledge mobilisation interventions alter and enhance mindlines to improve childhood eczema care? A United Kingdom based Social Impact Framework evaluation.
AUTHORS	Cowdell, Fiona; Lax, Stephanie; Van Onselen, Julie; Pendleton, Rose

VERSION 1 – REVIEW

REVIEWER	D'Auria, Enza Universita degli Studi di Milano, Pediatrics Department-Ospedale dei Bambini V. Buzzi
REVIEW RETURNED	22-Oct-2022

GENERAL COMMENTS	The paper aims to specifically evaluate the impact of co-created knowledge mobilisation interventions. The topic of the paper is interesting, however the initial enthusiasm is quite hampered after reading the paper. My comments as following: -In my opinion, the main limitation consists of the fact that all changes cannot be directly attributed to the KMB intervention. This hamper to draw firm conclusions. A critical comment by the authors should be appreciable-The declared aim of the paper was to described the impact of KMB activities. However, reading throughout the Text the impact of the interventions were performed (data source and data collection methods quite clearly detailed in table2) are not fully clear to the readersPoint 5 outcome 1 “you know your child’s eczema best” was not recorded: this crucial point is missingThe authors state that personal approaches may be more impactful, but Direct views to the video guides were n=121, as well as Uptake on Facebook and Twitter that achieved nine and 121 followers respectivelyCould they give a comment on these results?-Point 2:The implication from teacher feedback was that other children developed greater empathy for peers with eczema. It is not clear how this derives from teachers using Dragon resources. Could give a better explanation?-Discussion section is too much long and difficult to follow. It should be shortened, focusing on few messages the authors wish to give to the readers. Furthermore, the limitations of the study should be addressed (also in the Box strengths and limitations) and a more critical point of view by the authors should be explicated
---

REVIEWER	Swaithes , Laura Versus Arthritis Primary Care Centre, School of Medicine, Impact Accelerator Unit
REVIEW RETURNED	26-Oct-2022

GENERAL COMMENTS	This is a well written and interesting manuscript. The topic is contemporary, as many staff within the field are grappling with the challenges of measuring the impact of KM approaches. This manuscript raises many important points that knowledge mobilisers, knowledge brokers, researchers and clinicians may find useful. There are some components of the manuscript that I feel warrant clarification. Please see the questions and suggestions below that the authors may find helpful in enhancing their manuscript.  • The use of tables, figures, and illustrations is excellent and really helps to describe the study. Figure 1 needs reviewing for potential typos (for example under outcome 2 box) • The methods section would be strengthened by further detail and description on the data analysis process. Currently, it is not clear how the different data sources were 'analysed together'. How did the SIF guide the analysis? Were data sources combined, if so, how? What steps were taken to ensure that the analysis was robust? The reader may find it useful to have an overview or examples of the stages of analysis. • PPIE – you mention that lay people were involved at several stages of the study. Can you give any more detail as to the methods used to involve public contributors and the impact or outcomes that PPIE had in the study? Were there any challenges involving public contributors in this work? • The reader may find it helpful to have further clarification and detail relating to the role of the knowledge broker. This is mentioned briefly in Table 3 and in the results (outcome 4) but for those not familiar with the term/role it may be useful to provide additional information throughout the paper. For example, a definition of a knowledge broker, an example of who the knowledge brokers were in this study and who may act as a knowledge broker in other studies, explain how others took up the role of knowledge brokers, and explain more about the building of relationships for effective knowledge brokering. This seems like an important focus of this work which others may like to draw upon in other work. • It would be helpful to understand how KM activity and impact have been evaluated in other studies.
---

VERSION 1 – AUTHOR RESPONSE

Reviewer: 1 Dr. Enza D'Auria	
The paper aims to specifically evaluate the impact of co-created knowledge mobilisation interventions. The topic of the paper is interesting, however the initial enthusiasm is quite hampered after reading the paper. My comments as following: -In my opinion, the main limitation consists of the fact that all changes cannot be directly attributed to the KMb intervention. This hamper to draw firm conclusions. A critical comment by the authors should be appreciable	Thank you for your thoughtful comments. I am sorry that your initial enthusiasm was hampered. To address your point about attribution I have  • added to the explanation of contribution versus attribution on page • added a statement about contribution in the limitation section in discussion and in the strengths and limitations bullet points

The declared aim of the paper was to described the impact of KMB activities. However, reading throughout the Text the impact of the interventions were performed (data source and data collection methods quite clearly detailed in table2) are not fully clear to the readers Point 5 outcome 1 “you know your child’s eczema best” was not recorded: this crucial point is missing The authors state that personal approaches may be more impactful, but Direct views to the video guides were n=121, as well as Uptake on Facebook and Twitter that achieved nine and 121 followers respectively Could they give a comment on these results?	Thank you for raising this point. I have changed this to 23 on manuscript. It is a good point that there was some uptake of virtual interventions but this seems modest (given the potential audience size) when compared with for example 94 consultations in one shopping centre in one day. I have amended manuscript on page 10 “At the time of writing engagement with the mindlines webpage, Facebook and Twitter is modest particularly given the vast potential audience numbers (Table 6); this suggests that personal approaches may be more valuable”.
Point 2: The implication from teacher feedback was that other children developed greater empathy for peers with eczema. It is not clear how this derives from teachers using Dragon resources. Could give a better explanation?	I have added an illustrative quote on page 11 One teacher reported “we had a lot of discussion around the topic of eczema and talked about feelings and how things such as eczema can affect our moods”.
Discussion section is too much long and difficult to follow. It should be shortened, focusing on few messages the authors wish to give to the readers. Furthermore, the limitations of the study should be addressed (also in the Box strengths and limitations) and a more critical point of view by the authors should be explicated	Thank you, on reflection this section is rather long and I have rewritten to be more concise and convey key messages. As above, I have also amended bullet points
Reviewer: 2 Laura Swaithe	
This is a well written and interesting manuscript. The topic is contemporary, as many staff within the field are grappling with the challenges of measuring the impact of KM approaches. This manuscript raises many important points that knowledge mobilisers, knowledge brokers, researchers and clinicians may find useful. There are some components of the manuscript that I feel warrant clarification. Please see the questions and suggestions below that the authors may find helpful in enhancing their manuscript.	Thank you for your positive comments. I hope this paper will be of use to knowledge mobilisers and the wider health community.
The use of tables, figures, and illustrations is excellent and really helps to describe the study. Figure 1 needs reviewing for potential typos (for example under outcome 2 box)	Thank you, I am pleased that you liked the use of tables, figures and illustrations. I have corrected typo in Figure 1.
The methods section would be strengthened by further detail and description on the data analysis process. Currently, it is not clear how	Thank you, I have added detail about our analysis process but have tried to keep this succinct.

the different data sources were 'analysed together'. How did the SIF guide the analysis? Were data sources combined, if so, how? What steps were taken to ensure that the analysis was robust? The reader may find it useful to have an overview or examples of the stages of analysis.	
PPIE – you mention that lay people were involved at several stages of the study. Can you give any more detail as to the methods used to involve public contributors and the impact or outcomes that PPIE had in the study? Were there any challenges involving public contributors in this work?	I have added a little more detail to the PPI section. There were no challenges to their engagement that warrant reporting here.
The reader may find it helpful to have further clarification and detail relating to the role of the knowledge broker. This is mentioned briefly in Table 3 and in the results (outcome 4) but for those not familiar with the term/role it may be useful to provide additional information throughout the paper. For example, a definition of a knowledge broker, an example of who the knowledge brokers were in this study and who may act as a knowledge broker in other studies, explain how others took up the role of knowledge brokers, and explain more about the building of relationships for effective knowledge brokering. This seems like an important focus of this work which others may like to draw upon in other work.	Thank you, this is a good point, I've added a definition and pointed readers to further detail in Outcome 3.
It would be helpful to understand how KM activity and impact have been evaluated in other studies.	I have added a brief statement about this – the real answer is that we still have a lot of work to do but I think this example of using the SIF offers new insights and possibilities.

VERSION 2 – REVIEW

REVIEWER	Swaithes , Laura Versus Arthritis Primary Care Centre, School of Medicine, Impact Accelerator Unit
REVIEW RETURNED	03-Mar-2023
GENERAL COMMENTS	Thank you for addressing the comments thoroughly, in particular re the role of the knowledge broker - I think this is a valuable addition to the manuscript that will help others in the field